# Highly Sensitive ZnO/Au Nanosquare Arrays Electrode for Glucose Biosensing by Electrochemical and Optical Detection

**DOI:** 10.3390/molecules28020617

**Published:** 2023-01-07

**Authors:** Vinda Zakiyatuz Zulfa, Nasori Nasori, Ulya Farahdina, Miftakhul Firdhaus, Ihwanul Aziz, Hari Suprihatin, Muslikha Nourma Rhomadhoni, Agus Rubiyanto

**Affiliations:** 1Laboratory Medical Physics and Biophysics, Department of Physics, Faculty of Sciences and Data Analytic, Sepuluh Nopember Technology Institute, Surabaya 60111, Indonesia; 2Research Center for Accelerator Technology, Research Organization of Nuclear Energy, National Research and Innovation Agency (BRIN), Yogyakarta 55281, Indonesia; 3Occupational and Safety Department, Nahdlatul Ulama University of Surabaya, Surabaya 60237, Indonesia

**Keywords:** zinc oxide, nanosquare arrays, glucose, electrode-sensing, biomedics

## Abstract

The fabrication of a ZnO/Au nanosquare-array electrode was successfully carried out for the detection of glucose concentration in biomedical applications. The fabrication of the ZnO/Au nanosquare array using an ultra-thin alumina mask (UTAM) based on the imprinted anodic aluminum oxide (AAO) template and the direct current (DC) sputtering method was able to produce a very well-ordered nanosquare arrangement with a side size of 300 nm and a thickness of 100 nm. Tests were done to evaluate the performance of the electrode by means of cyclic voltammetry (CV) which showed that the addition of glucose oxidase (GOx) increased the sensitivity of the electrode up to 1180 ± 116 μA mM^−1^cm^−2^, compared with its sensitivity prior to the addition of GOx of 188.34 ± 18.70 mA mM^−1^ cm^−2^. A i_ox_/i_red_ ratio equal to ~1 between the peaks of redox reactions was obtained for high (hyperglycemia), normal, and low (hypoglycemia) levels of glucose. The ZnO/Au nanosquare-array electrode was 7.54% more sensitive than the ZnO/Au thin-film electrode. Furthermore, finite-difference time-domain (FDTD) simulations and theoretical calculations of the energy density of the electric and magnetic fields produced by the ZnO/Au electrode were carried out and compared to the results of CV. From the results of CV, FDTD simulation, and theoretical calculations, it was confirmed that the ZnO/Au nanosquare array possessed a significant optical absorption and that the quantum effect from the nanosquare array resulted in a higher sensitivity than the thin film.

## 1. Introduction

Impaired glucose metabolism and high levels of glucose concentration are the main indicators of diabetes mellitus (DM) [1]. The normal level, low level (hypoglycemia), and high level (hyperglycemia) of glucose concentration in blood is in the range of 100 mg/mL, lower than 75 mg/mL, and higher than 130 mg/mL, respectively [2]. Furthermore, DM can also be indicated by insulin resistance due to uncontrolled levels of calcium in the body [1,3]. To identify these indicators of DM, biosensor technology provides a solution for the detection and monitoring of analytes such as glucose, enzymes, antibodies, proteins, DNA, and RNA for clinical analysis and medical diagnosis. The three components of biosensors that affect its sensitivity are the bioreceptor, transducer, and detector [4].

In recent decades, nanostructures have had a significant impact on biosensor applications, particularly in clinical diagnosis to monitor human health [5]. Biosensors based on nanostructure arrays can provide satisfactory analysis results because they possess a high surface area and a structure depth that is in contact with the target molecule which, as a result, produces analytical signals from its measurement [6]. Recent research on biosensors based on nanostructure arrays have shown that they can also provide high sensitivity and selectivity, and are easy to apply for daily use [7,8]. Zinc oxide (ZnO) is a metal oxide that has a high potential to be used as a biosensor due to its high stability, small band-gap value, and relatively low price [9]. In a previous study, it was shown that ZnO possesses an isoelectric point (IP) of around ±9.5; therefore, this metal is also suitable for protein absorption. Glucose oxidase (*G_ox_*) can also be added to the electrode to catalyze the oxidation of β-d glucose in glucose biosensors [10]. However, the low conductivity of ZnO causes the formation of free charges in a relatively short period of time. Therefore, the electron transfer in the redox reaction that occurs must be devoid of obstruction as much as possible and take place continuously from the enzyme to the electrode [11]. For this reason, researchers in the field of biosensors are very critical in selecting metal oxides such as Ti_2_O and ZnO as electrode materials [8,9,10]. In order to accelerate the oxidation of glucose, it is common to add other components such as Au nanoparticles, carbon nanotubes, and CdS quantum dots [11,12,13,14]. Furthermore, several researchers have utilized biomaterial substrates such as egg shell and bamboo inner-shell membranes as an effective and stable enzyme immobilization platform.

On the basis of previous research, we used an ultra-thin alumina mask (UTAM) based on the anodic aluminum oxide (AAO) template as a fabrication method for nanostructures. This method has been shown to successfully fabricate ZnO nanostructures with varying morphologies such as nanotubes and nanorods [6,15,16]. By chemically controlling the scale of the fabrication process of the AAO template, a homogeneous porous nanostructure and the desired surface-to-volume ratio can be achieved. This method is relatively low-cost because it uses a chemical–physical process that is safe in its preparation stage. Furthermore, there are many non-nanomaterials that can be fabricated into nanostructures, for instance by using the DC sputtering deposition method [17]. In this study, we fabricated ZnO nanosquare arrays on a fluorine-doped tin oxide (FTO) substrate that was thinly coated with Au. Important aspects in the field of biosensors and applications of optical sensors, particularly the optical band gap, are the main focus of this study. Simulations using finite-difference time-domain (FDTD) and mathematical calculations support the fact that ZnO/Au nanosquare arrays have a significant effect on the absorption pattern without considering the quantum effect. Therefore, this paper provides a new outlook for looking at the shift in absorption onset of semiconductors. Compared with recent studies, this research includes the simulation of the distribution of the electric field that can be used as the basis for making biosensors. In addition, this research manufactures nanostructure-mode biosensors with a fairly simple method which can therefore also be developed as paper-like biosensors because most of the very thin biosensors are still in the form of nanoparticles arranged as thin layers [18].

## 2. Results and Discussion

SEM images of the top view and side view of the ZnO nanosquare fabricated using the imprinted AAO template and deposited on top of a thin Au layer is shown in Figure 1a. From the images, it can be clearly seen that the resulting nanostructure shape was a nanosquare array. The ZnO/Au nanosquare was used as the model to determine the energy density of the electric field generated by its structure. When considering other optical properties, the geometric shape of the ZnO nanosquare array plays a crucial role in precisely determining the contents of the deposited atoms and as well as identifying whether changes have occurred after the removal of the AAO template using sodium sulfide. By carrying out these processes on the ZnO nanosquare array, it can be proven that a good structure can be obtained and also provide a description of the optical properties and geometric structure, which are the main focus of this study.

As previously described, other chemical compounds were involved in the fabrication process of the ZnO nanosquare array. Figure 1b shows the XRD pattern after the removal of the AAO template from the substrate; the diffraction peaks of ZnO were at 31.7° 2θ, 34.5° 2θ, and 36.2° 2θ, corresponding to the (100), (002), and (101) planes, respectively, based on the ZnO wurtzite (JCPDS No. 005-0664), where the lattice constant *a* = 0.32 nm and *c* = 0.52 nm. The diffraction peak corresponding to the (002) plane was smaller compared to that corresponding to the (100) and (101) planes. Diffraction peaks of Au that were placed underneath were at 38.2° 2θ, 44.3° 2θ and 78.4° 2θ, corresponding to the (111), (112), and (311) planes, respectively, according to the standard Au pattern (JCPDS No. 04-07072). These peaks were also obtained in the study conducted by Fatemi et.al, who used a different method in the application of glucose biosensors [12].

Figure 1c–e shows the mapping of the elements that comprised the material from the SEM/EDX analysis, in which all the materials that comprised the electrode, namely O, Au, and Zn, can be clearly seen in detail. In Figure 1d,e, the geometry and depth of the elements can be seen and provides the opportunity to explain the effect of the large surface area that is connected to the medium, namely the glucose solution.

The performance of the ZnO/Au nanosquare array and thin film in detecting glucose concentration was electrochemically tested using CV in [Fe(CN)_6_]^3−/4−^ and glucose media with varying concentrations and scan rates. The cyclic voltammetry data displayed is the scan result in the fifth cycle because, in that cycle, the results of the cyclic voltammetry reading have reached stability. Prior to the testing using CV, the solution was flowed with nitrogen to reduce bubbles and maintain the purity of the solution. The first testing of the specimen was carried out using a mixed solution of [Fe(CN)_6_]^3−/4−^ 1.0 mM in a solution of 0.1 M KCl. The CV curves shown in Figure 2a,b exhibit several interesting information for evaluation. Firstly, the CV patterns and peaks of reduction and oxidation in the [Fe(CN)_6_]^3−/4−^ solution that occur on the surface of the ZnO/Au thin film and ZnO/Au nanosquare array are quite different to one another. By increasing the concentration from 0.12 mM to 0.18 mM that was scanned with a scan rate of 0.12 V s^−1^, an increase in peak currents of reduction and oxidation occurred. This is in accordance with the Randles–Sevcik equation that states that the concentration of the solution is proportional to the oxidation and reduction currents. A shift in the position of peak potential vs. current is made possible due to the structure formed by the two electrodes, resulting in a larger increase in electrochemical response in the oxidation reaction compared to the reduction reaction. Therefore, the CV curve of the ZnO/Au thin-film electrode appears to be uneven with a peak oxidation current of approximately 0.0158 mA/cm^2^, while the CV curve of the ZnO/Au nanosquare array appears to be more even with a peak oxidation current of 0.266 mA/cm^2^. As shown in Figure 2a,b, the results of the scan using the ZnO/Au nanosquare array experienced a significant shift compared to the scan using the ZnO/Au thin film due to the larger surface area-to-volume ratio of the nanosquare array, which increases the possibility of the occurrence of a redox reaction on the surface of the electrode [19]. This causes the ZnO/Au nanosquare-array electrode to be more sensitive compared to the ZnO/Au thin film. The sensitivity value can be obtained from the gradient of the graph shown in Figure 2c. It can be concluded that the ZnO/Au nanosquare array is 7.54% more sensitive than the ZnO/Au thin film. The sensitivity of the ZnO/Au thin film and ZnO/Au nanosquare array was 0.97 μA mM^−1^cm^−2^ and 1.05 μA mM^−1^cm^−2^, respectively.

Theoretically, the standard deviation can be calculated using the Laviron equation as follows:log *ks* = α log (1 − α) − log (RT/nFυ) − α(1 − α) nFΔE_p_/2.3RT(1)
where α is the standard deviation of current density, n is the number of electrons transferred (n = 1), F is the Faraday constant (96,485 C mol^−1^), and ∆E_p_ is the potential difference [16,17,20]. From the calculations of standard deviation, the limit of detection (LOD) with varying scan rates (between 0.12 V s^−1^ and 0.2 V s^−1^) and varying concentrations as shown in Figure 2a,b can be calculated. The obtained LOD value for the ZnO/Au nanosquare array and ZnO/Au thin film was 0.0877 mM and 0.0931 mM, respectively.

Figure 3 shows the kinetics of electrode reactions of ZnO/Au electrodes in the form of thin film and nanostructure. The results of CV show a significant change in the CV curves of both electrodes at a glucose concentration of 0.2 mM. In the reduction and oxidation reaction that occurred in the CV test, the peak of oxidation potential for the ZnO/Au thin film and ZnO/Au nanosquare was 0.4 V vs. Ag/AgCl and 0.55 V vs. Ag/AgCl, respectively, at varying scan rates (from 0.12 V s^−1^ up to 0.2 V s^−1^ vs. Ag/AgCl). There exists a linear relationship between the peak current and the scan rate. This can be proven by an increase in the peak oxidation and peak reduction currents with respect to an increase in scan rate as shown in Figure 3b,d. This indicates that oxidation and reduction reactions possess a unique pattern, and also that diffusion in the electrochemical process was well-controlled by the electrode. On the other hand, the peak cathodic current i_pca_ and the peak anodic current i_pan_ experienced a slight shift in the peak of the reduction reaction.

The results of the experiment on the effect of scan rate on the peak anodic current and peak cathodic current were in accordance to the Randles–Sevcik theory, in which it is stated that the peak anodic and cathodic current values is linearly proportional to the scan rate. A comparison between the oxidation current (i_ox_) and reduction current (i_red_) shown in Figure 3b,d shows that the i_ox_/i_red_ was slightly bigger than 1. This indicates that chemical reactions rarely occurred on the ZnO/Au thin film electrode; furthermore, the transfer of charges can occur reversibly on both the ZnO/Au thin-film and ZnO/Au nanosquare-array electrodes.

Figure 4 shows the sensitivity stability of both the ZnO/Au thin-film and ZnO/Au nanosquare-array electrodes in detecting glucose. A reduction in the concentration of glucose in the medium by 0.18 mM was carried out every 500 s by means of adding solvents to the medium. A decrease in the concentration of glucose will reduce the ability of electroactive substances to undergo a change in oxidation state; therefore, a decrease in oxidation and reduction currents occur. Based on the results shown in Figure 4, the ZnO/Au nanosquare array possesses a standard deviation of 1.4% for every reduction of glucose concentration, which is smaller than the standard deviation of the ZnO/Au thin film (2.2%). Therefore, the sensitivity of the ZnO/Au nanosquare array was more stable than that of the ZnO/Au thin film. Moreover, Figure 4 also shows that the current response of ZnO/Au nanosquare electrodes to light with a wavelength of 450 nm is better when compared to thin films. The test results can be used as the basis to conclude that the ZnO/Au nanosquare electrode is better than thin film.

In biosensor applications, Au can act as a substrate conductivity enhancer and molecular bonds with electrodes. Therefore, adding a thin Au layer which has low ohmic properties to chemical response beneath the ZnO nanosquare array can significantly affect the sensitivity of the electrode due to the high biocompatibility and conductivity of Au. The addition of a thin Au layer was shown to increase the sensitivity of ZnO thin films to detect glucose [21]. In the application of this ZnO nanosquare electrode, the presence of a thin Au layer on top of the FTO substrate functions as a Schottky barrier between FTO and the ZnO nanosquare array. As a result, the conduction band and the transfer of hot electrons to the ZnO nanosquare becomes lower [22]. Furthermore, Au can speed up the transfer of holes as a positive charge in the photogeneration process. In addition, the thin Au layer is able to block the return of holes to the valence band at the interface between the thin Au layer and the ZnO nanosquare array [23].

The anodic peak potential and the cathodic peak potential for the ZnO/Au thin film with the addition of GOx, shown by the red curve in Figure 5, appears to be different from how it should be. The use of an electroactive substance as the medium should result in a controlled pattern. In the case of this research, the media used were glucose and [Fe(CN)_6_]^3−/4−^. Using a scan rate of 0.05 V s^−1^ and the medium of 5.0 mM [Fe(CN)_6_]^3−/4−^ and 0.2 mM of glucose for the ZnO/Au thin film, the potential difference (Δ*Ep*) increased with respect to an increase in concentration until a value of 0.820 V was achieved, while the optimal value of potential difference for the ZnO/Au nanosquare array was 0.114 V at the same scan rate and concentration, as shown on Figure 5. The Δ*Ep* value for the ZnO/Au nanosquare array was lower to that of the ZnO/Au thin film which indicates that the electron transfer kinetics was faster with the ZnO/Au nanosquare array, particularly with the use of Gox at the stabilized electrode interface. Due to the fact that electron transfer kinetics do not depend on the concentration of the electroactive compound, the effect of resistance does not accumulate and is not dependent on the concentration. However, in a previous study it was stated that the increase of ∆*Ep* that is dependent on the concentration may be caused by the effect of resistance that has not been compensated [24]. To clarify this relationship, Nicholson used the heterogeneous electron transfer rate constant (*ks*) from [Fe(CN)_6_]^3−/4−^ that evaluates certain changes in concentration and scan rate [25].

The results of the experiment and analysis conducted in this study showed that the nanostructure possessed a higher sensitivity compared to the thin film. The sensitivity of the ZnO/Au electrodes in detecting glucose was tested by means of CV using a glucose medium with the addition of GOx at 2.5 × 10^−5^ mM. CV was conducted with an effective scan rate of 0.12 V s^−1^. For the two types of electrodes, namely thin-film and nanosquare-array, there was a significant difference between the CV curves of both electrodes, as shown in Figure 5a,c, which indicated that the response of the ZnO/Au nanosquare array was higher than that of the ZnO/Au thin film. With the addition of GOx to the solution, a peak oxidation current and peak reduction current of 0.485 V and 0.344 V, respectively, was obtained for the ZnO/Au nanosquare array, compared with a peak oxidation current and peak reduction current of 0.478 V and 0.340 V, respectively, for the ZnO/Au thin film. Furthermore, an increase in I_pca_ and I_pan_ as shown in Figure 5b,d indicates faster electron transfer kinetics in the active cycles of GOx. The larger increase in peak oxidation and reduction currents for the ZnO nanosquare is also caused by the larger surface area of the ZnO/Au nanosquare which contributes to the electrical conductivity for the electron transfer in flavin adenine dinucleotide (FAD) compounds and its derivatives such as FADH_2_. These compounds play an active role as catalysts in metabolism; GOx reacts with FADH_2_ that possesses two negative charges and two positively charged hydrogens; consequently, more GOx and FADH2 will be formed and interact with a larger electrode surface. The oxidation and reduction reactions that may occur on the electrode and the solution are as follows:Zn^2+^ + 2OH^−^ →Zn(OH)_2_(2)
Zn(OH)_2_ + 2OH^−^ ↔ [Zn(OH)_4_]^2−^(3)
Glucose + GOx → Gluconolactone +H_2_O_2_(4)

The oxidation and reduction peaks at a potential of 0 V indicate that there is an oxidation and a reduction reaction in glucose, while the peaks of oxidation and reduction at 0.45 V and 0.2 V indicate the presence of oxidation and reduction reactions in Zn [26,27,28]. The results of CV proved that the fabricated ZnO/Au nanosquare electrode with the addition of GOx had a higher sensitivity, namely 1180 ± 116 μA mM^−1^cm^−2^, compared to the electrode without the addition of GOx, as shown in Figure 6b. From the experiment that was conducted, the obtained LOD value for the ZnO/Au nanosquare and thin film with the addition of GOx was 0.0616. Compared to the electrode without the addition of GOx, the electrode with the addition of GOx had a higher sensitivity and a lower LOD.

The selectivity of the GOx/ZnO/Au nanosquare array sensors was tested with three different types of analytes. In addition to using glucose, evaluation was also carried out using fructose and sucrose. CV testing for fructose and sucrose is shown in Appendix A. The results of the selectivity test identified by an increase in the oxidation current are shown in Figure 6c with a change in concentration between 0.12 mM to 0.20 mM. Evaluation by glucose analytes produced a superior change in current for the increase in glucose compared to other types of analytes because the interaction of glucose with GOx can release electrons, thereby increasing the electric current significantly [26]. In contrast, the ZnO/Au nanosquare array biosensor tested with sucrose and fructose analytes did not produce a significant increase in current because it has a large enough anti-interference ability which can be related to physical isolation and electrode selectivity [27].

In accordance with the initial aim of this study, analysis of the optical properties of the material was carried out for electrochemical observations using FDTD simulations and a model similar to the results of SEM shown in Figure 1a. Electric field distribution was simulated on the Au thin film and ZnO/Au nanosquare with a side size of 300 nm. The ZnO nanosquare was placed on top of the Au thin film on a glass substrate. The electric field-distribution simulation was carried out using electromagnetic waves with wavelengths in the range of 300 up to 1000 nm. The top part of the ZnO nanosquare was modeled by coating using normal, hypoglycemia, and hyperglycemia blood media. The electric field-distribution simulation was based on the Gauss and Ampere laws; therefore, the electric field on each component of the electrode was very dependent on the permittivity and permeability values [29].

Table 1 shows the sensitivity values of glucose measurement compared to existing studies. Various forms of nanostructures were used in previous studies such as nanorods, nanowires, and urchin-like nanostructures. However, the results show that ZnO, which has a high nanostructure, is more sensitive than the others. In this work, we obtained a high sensitivity value for glucose determination compared to previous research.

The electric field distribution on the ZnO electrode is shown in Figure 7. The electromagnetic wave was perpendicularly irradiated on top of the medium, which resulted in a relatively high electric field on the top part of the medium. The electromagnetic wave was gradually distributed through the medium, ZnO nanosquare, Au thin film, and glass substrate. The electric field distribution on each component of the electrode possessed different values due to different characteristics in passing on, reflecting, and absorbing the energy from the electromagnetic wave. The electric field in the medium was larger due to its relatively smaller permittivity compared to the other components of the electrode. Based on the simulation, despite the fact that the electric field on the ZnO electrode possessed a higher intensity, namely 450 nm, the electric field was larger, reaching a value of 6.12 × 10^7^ V/m around the ZnO electrode when the nanosquare was irradiated with electromagnetic waves with a wavelength of 550 nm. This high electric field value can cause molecules to move towards that area. 

Dissimilar glucose concentrations in blood can indicate the ability of the body to regulate blood glucose [34]. An increase in glucose concentration can increase the refractive index of blood [35]. A change in refractive index of the blood medium due to a change in glucose concentration may in turn cause a change in the electric field distribution on the ZnO nanosquare. Figure 8 shows the electric field distribution on the ZnO nanosquare that was irradiated with electromagnetic waves with a wavelength of 550 nm in varying media. The maximum electric field value in the hypoglycemia, normal, and hyperglycemia blood media were 2.98 × 10^7^ V/m, 4.18 × 10^7^ V/m, and 5.78 × 10^7^ V/m, respectively. The results of the simulation show that there is an increase in the electric field value on the ZnO nanosquare with respect to an increase in blood glucose concentration, and so this can be used as a parameter to detect the level of glucose in blood using the ZnO nanosquare electrode.

The maximum electric field distribution can also be identified from the total electric and magnetic energy on the ZnO nanosquare. Figure 9 shows the total electric and magnetic value of the ZnO nanosquare that was irradiated with electromagnetic waves with wavelengths in the range of 300 up to 1000 nm. The maximum electric energy and magnetic energy was obtained in the hyperglycemia blood medium, in which the maximum electric energy and magnetic energy obtained was 9.43 × 10^−16^ Joule and 9.90 × 10^−16^ Joule, respectively. The maximum electric and magnetic energy value was obtained when the nanosquare with a side size of 300 nm was irradiated with electromagnetic waves with a wavelength of 450 nm. Maximum electric and magnetic energy values were caused by a higher interaction with light in the hyperglycemia blood medium. Therefore, these simulation results on the electric and magnetic energy can be used as the optimal parameter for the fabrication of electrodes to detect hyperglycemia.

The transmittance of the ZnO nanosquare with varying wavelengths and media is shown in Figure 10. The peak transmittance values in the hypoglycemia, normal, and hyperglycemia blood media were 0.512, 0.517, and 0.510 a.u, respectively. The transmittance was higher with respect to a decrease in the concentration of the medium; this is caused by an increase in optical density of the media due to an increase in glucose concentration [36]. The optical density value is inversely proportional to the transmittance value [37]. The transmittance of the ZnO nanosquare differs in blood media with low, high, and normal glucose concentrations due to the different refractive index and electric field distribution. Furthermore, the study conducted by Webster et al. in 2009 stated that each molecule possesses a rotational oscillation and vibration that differs from one another and, as a result, possesses different resonance absorption peaks [38]. In addition, the shift in the transmittance peak can also be caused by changes in the index of refraction due to the bond formed between the analyte and the electrode [39]. The simulation results show that the peak transmittance in the hypoglycemia, normal, and hyperglycemia blood media were obtained with a wavelength of 500 nm, 500 nm, and 550 nm, respectively. The shift in peak and intensity of transmittance with different wavelengths of the ZnO/Au nanosquare in blood media with varying glucose concentrations can be used as the basis to detect glucose concentration in blood using spectroscopic methods.

The transmittance values were calculated based on the ratio between the total electric field intensity on the top part of the medium and the bottom part of the Au thin film [40]. The transmittance of the ZnO nanosquare in different blood media with varying glucose concentrations are shown in Figure 10. Based on these results, it can be seen that the peak transmittance patterns from the simulation and calculation were similar, in which the peak transmittance value increased and the wavelength decreased with respect to a decrease in glucose concentration in blood. The results show that peak transmittance in the hypoglycemia, normal, and hyperglycemia blood media was 0.558 a.u at a wavelength of 550 nm, 0.744 a.u at a wavelength of 550 nm, and 1.028 a.u at a wavelength of 500 nm, respectively. Based on these results, the transmittance value obtained from both the simulation and calculation can be used to detect glucose concentration in blood using spectroscopic methods by identifying the peak transmittance values and their corresponding wavelengths.

## 3. Materials and Methods

### 3.1. Materials

Si wafer (99%), nickel(II) sulfate (NiSO4; 98%), nickel(II) chloride (NiCl_2_; 98%), boric acid (H_3_BO_3_; 97%), sodium hydroxide (NaOH; 99%), titanium tetrachloride (TiCl_4_; 99%), phosphoric acid (H_3_PO_4_; 70%), sodium sulfate (Na_2_SO_4_; ≥99%), copper(II) chloride (CuCl_2_; 98%), acetone (C_3_H_6_O; 99.8%), polymethyl methacrylate (PMMA; (C_5_O_2_H_8_)n), and granular ZnO (99%) were utilized as obtained from Sigma Aldrich (Singapore, Singapore). The Loctite 3430 insulating epoxy was used. Thick Al foil (99.99%) and Copper Cu tape (AT528) were acquired from Advance Tapes (Surabaya, Indonesia), while the TEC-15 FTO glass (density 150 nm/1.1 mm; resistivity of sheet ≤ 20 W/cm^2^, transmissivity 88.9%; wavelength 400 nm) was bought from NSG Glass(Tokyo, Japan).

### 3.2. Fabrication of Ultra-Thin Alumina Masks, Au Thin Films, and ZnO Nanosquare Arrays

The technology referred to as UTAM is a template that is made by using AAO based chemical compounds. The UTAM was made from aluminum (Al) with a purity of 99.99% and thickness of 0.22 ± 0.005 mm. Al that was shaped to be rounded with a diameter of 3 ± 0.5 cm was washed with acetone (99%) and deionized (DI) water for 30 ± 0.5 min in an ultrasonic bath. After it was cleaned, the surface of Al was electrochemically polished using HClO_4_ and ethanol in a ratio of 1:7 and an applied voltage of 30 V for 2 min. Concurrently, the fabrication of a nickel (Ni) stamp on a silicon (Si) wafer was carried out by means of electrodeposition in a solution of nickel(II) sulfate (NiSO_4_) and nickel(II) chloride (NiCl_2_) with an applied current of 1 mA for 8 h. Subsequently, the surface of the Si wafer that had been deposited with Ni was peeled off which resulted in a shiny surface that had a color spectrum complexion. Afterwards, the Ni stamp was placed on top of the polished Al surface with a pressure of 3 kPa for 3 min.

The anodization of Al was carried out under a controlled temperature of 2 °C using a 180 V power supply for 20 min. To control the width and depth of the pores, the AAO specimen was submerged in a solution of H_3_PO_4_ (w 5%) for 2 h at temperatures between 18 to 20 °C. Next, the PMMA was adhered onto the surface of the imprinted AAO template, then dried at room temperature, which resulted in a specimen arrangement of Al/AAO/PMMA. To obtain the UTAM from the imprinted AAO template, Al had to be removed first by submerging in a CuCl_2_ solution. Subsequently, PMMA was removed by submerging in a HCl solution at room temperature to obtain the UTAM. To clean the UTAM, the specimen was submerged in a solution of acetone and DI water for 3 h. UTAM was transferred onto the surface of the FTO that had been deposited with a thin Au layer by using the e-beam physical vapor deposition (PVD) (Kurt J. Lester) method at a speed of 20 rpm which resulted in a 5 ± 0.5 nm layer of Au (step I). After the UTAM had been slowly and perfectly transferred onto FTO/Au, it was cleaned with DI water and then dried and placed on a dry tissue (step II). After it was completely dry, ZnO deposition was carried out using DC sputtering (CY-GZK103-A BRIN, Yogyakarta, Indonesia) with argon gas as the sputter on FTO/Au/UTAM (step III). Subsequently, the UTAM was removed by using a dry and clean tape and quickly pulled off. As a result, a ZnO nanosquare array and a thin Au layer directly below it was obtained (step IV) as shown in Figure 11.

### 3.3. Characterization and Performance of Electrodes

X-ray diffraction spectrum (XRD) with Ni-filtered Cu Kα radiation using a Siemens D500 Diffractometer(Institut Teknologi Sepuluh Nopember, Surabaya, Indonesia) was carried out to identify the peaks of the materials that composed the electrode at an angle of 0 to 2θ. Surface morphological analysis and mapping of the ZnO/Au nanosquare were carried out using scanning electron microscopy with energy dispersive X-ray spectroscopy (SEM/EDX) analysis using the Phenom ProX-G6(Dynatech International, Jakarta, Indonesia). Electrochemical reactivity of the ZnO/Au nanosquare and thin film were carried out using Corrtest and cyclic voltammetry (CV). The voltage interval was between −1 to 1 V vs. Ag/AgCl as the reference electrode and a Pt sheet as the counter electrode. Electrochemical tests were done in a solution of 1 mM [Fe(CN)_6_]^3−/4−^, G_ox_, and glucose with a concentration between 0.12 mM up to 0.18 mM. The scan rate was varied between 0.12 V s^−1^ up to 0.220 V s^−1^.

### 3.4. Simulation Methods

Three-dimensional FDTD simulations were carried out using the FDTD solutions program from Comsol Multiphysics software (Version 5.4 COMSOL Compiler™, Stockholm, Sweden). The smallest model grid size was used for the accuracy and stability of the FDTD calculation, which was obtained iteratively using convergence testing. Convergence testing was done by calculations before the simulation with a grid size of λ_0/20_, where λ_0_ is the minimum wavelength expected in the simulation; after that, the grid size was decreased by half in consecutive simulations.

## 4. Conclusions

In conclusion, we explained in detail the fabrication of ZnO/Au thin-film and ZnO/Au nanosquare-array electrodes and conducted experiments to investigate their application in glucose biosensors. We investigated the quantum effect of the ZnO/Au thin-film and nanosquare-array electrodes in the presence of photon interactions on the nanostructure, which provided significant knowledge on the conductivity and optical properties of semiconductors. The degree of linearity of the cathodic and anodic peak currents in the glucose solution increased to up to two times its initial value for the ZnO/Au nanosquare-array electrode, achieving a sensitivity of 188.34 ± 18.70 mA mM^−1^ cm^−2^ at a scan rate of 0.16 V s^−1^. By comparison, the ZnO/Au thin-film electrode achieved a sensitivity of 153.73 ± 15.22 mA mM^−1^ cm^−2^ at the same scan rate. The ZnO/Au nanosquare array achieved a higher sensitivity due to the larger surface-to-volume ratio compared to the ZnO/Au thin film. Modifications by adding a thin Au layer to the electrode can increase the chemical response of the electrode. The addition of GOx to the glucose solution can increase the sensitivity of the electrode up to 1000 times due to the increase in redox in glucose and water as a result of adding GOx. A suitable tunability allows ZnO/Au nanosquare arrays to have great potential in optical applications such as light absorption by shooting photons with wavelengths of 516 nm, 517 nm, and 548 nm for low blood glucose (hypoglycemia), normal blood glucose, and high blood glucose (hyperglycemia), respectively.

## Figures and Tables

**Figure 1 molecules-28-00617-f001:**
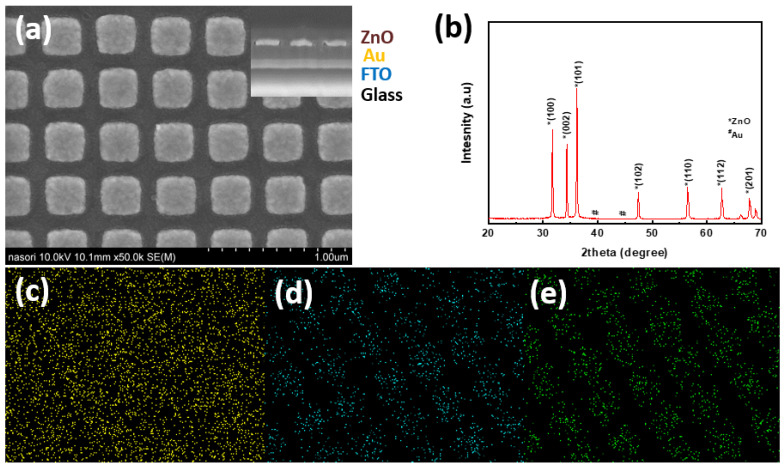
(**a**) SEM image of the top view of a square-shaped ZnO/Au array (inset: Cross-sectional SEM images of the ZnO/Au nanosquare array), (**b**) XRD patterns of the ZnO/Au representative of EDX mapping image for (**c**) Au element, (**d**) O element and (**e**) Zn element.

**Figure 2 molecules-28-00617-f002:**
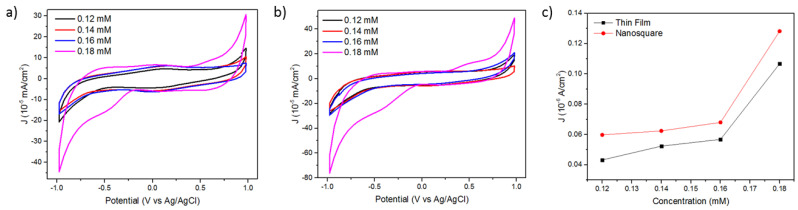
(**a**) CV curve of the ZnO/Au thin film in the [Fe(CN)_6_]^3−/4−^ solution and (**b**) CV curve of the ZnO/Au nanosquare array in the [Fe(CN)_6_]^3−/4−^ solution at various glucose concentrations from 0.12, 0.14, 0.16, 0.18 mM; and (**c**) the graph that depicts the linearity between peak currents of reduction and oxidation and the concentration with deviation standard of 0.005μA/cm^2^. The data was taken from the peak of the CV result on potential of 0.2 V vs. Ag/AgCl from the inner part of the curve up to the outer part.

**Figure 3 molecules-28-00617-f003:**
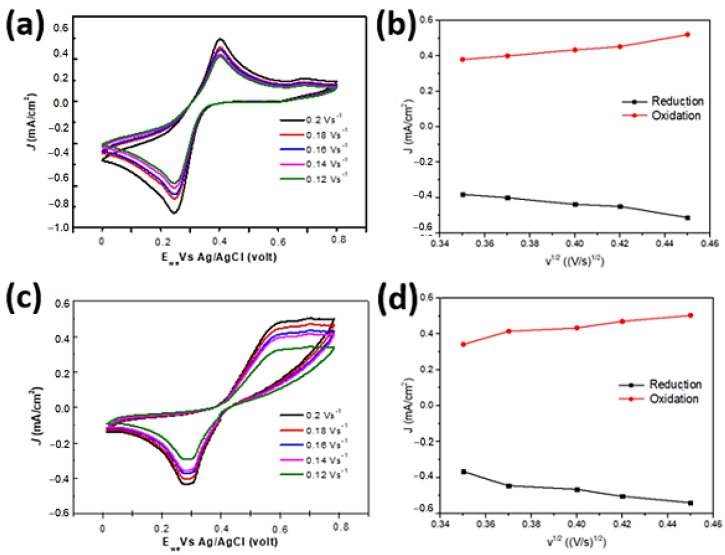
(**a**) CV curve of the ZnO/Au thin film in the 5 mM [Fe(CN)_6_] ^3−/4−^ solution and (**c**) CV curve of the ZnO/Au nanosquare array in 5.0 mM [Fe(CN)_6_]^3−/4−^ and 0.2 mM of glucose analyte medium at various scan rates between 0.02 V s^−1^ up to 0.12 V s^−1^. Graph that depicts the linearity between peak currents of reduction and oxidation at each scan rate for (**b**) ZnO/Au thin film and (**d**) ZnO/Au nanosquare array. Peak currents of oxidation (i_pox_) on the ZnO/Au thin film: 0.250 mA/cm^2^–0.538 mA/cm^2^, and peak currents of reduction (i_pred_): −0.199 mA/cm^2^–−0.421 mA/cm^2^. Peak currents of oxidation (i_pox_) on the ZnO/Au nanosquare array: 0.231 mA/cm^2^–0.311 mA/cm^2^, and peak currents of reduction (i_pred_): −0.245 mA/cm^2^–−0.353 mA/cm^2^ at a concentration of 9.90 × 10^−5^ mol L^−1^. The data were taken from the inner part of the curve up to the outer part.

**Figure 4 molecules-28-00617-f004:**
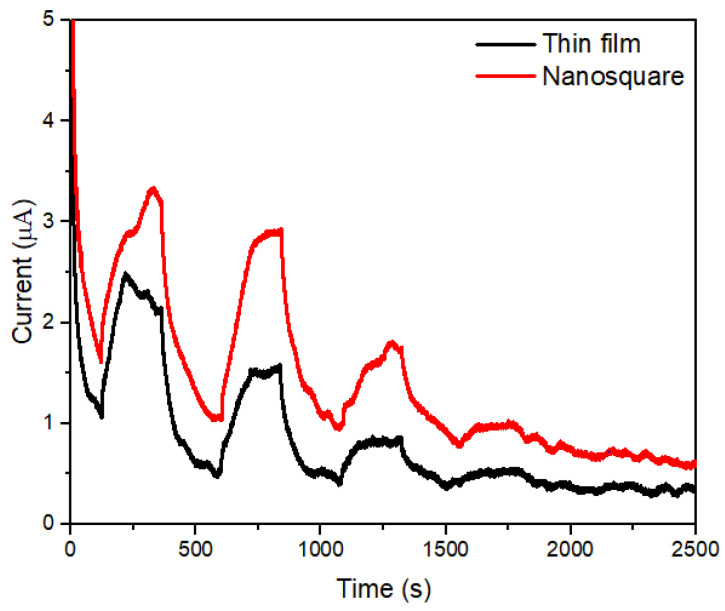
The stability curve of the ZnO/Au thin film and the ZnO/Au nanosquare arrays at scan rates between 0.12 V s^−1^ and 0.18 mM of glucose concentration under a 450 nm wavelength.

**Figure 5 molecules-28-00617-f005:**
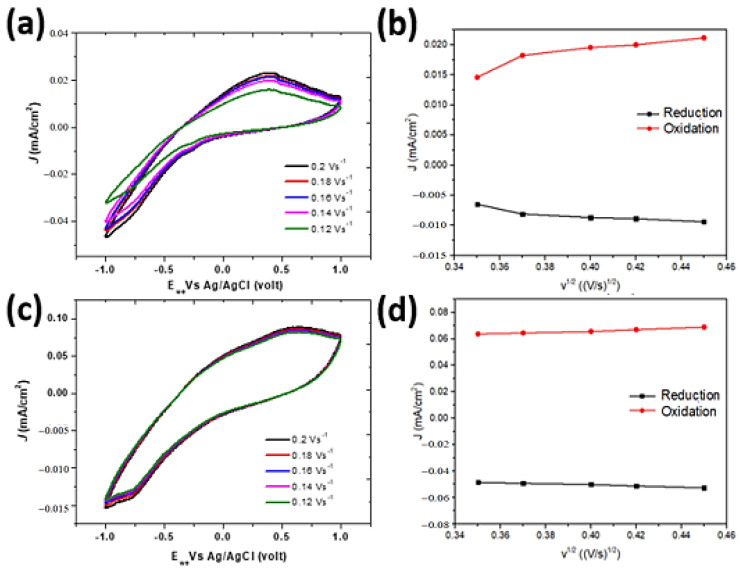
The CV curves of the electrode samples of (**a**) GOx/ZnO/Au thin film and (**c**) GOx/ZnO/Au nanosquare array with scan rates in the range of 0.02 V s^−1^ and 0.12 V s^−1^ in 5.0 mM [Fe(CN)_6_]^3−/4−^ and 0.2 mM of glucose analyte. Graph that represents peak oxidation currents (red line) and peak reduction currents (black line) for the (**b**) GOx/ZnO/Au thin film and (**d**) GOx/ZnO/Au nanosquare array.

**Figure 6 molecules-28-00617-f006:**
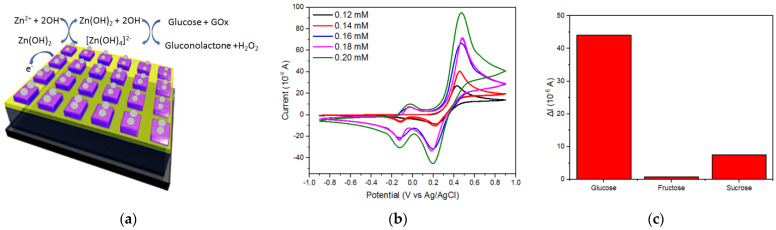
(**a**) Schematic of electrocatalytic reactions on the ZnO/Au electrode using Gox; (**b**) CV graph of the ZnO/Au nanosquare array with the addition of GOx with varying concentrations of glucose; and (**c**) selectivity of the GOx/ZnO/Au nanosquare array.

**Figure 7 molecules-28-00617-f007:**
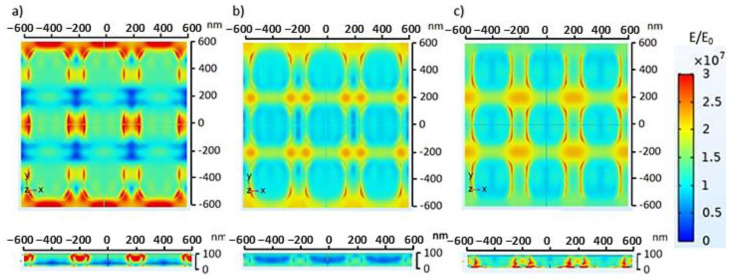
Top view and side view of the electric field distribution on the ZnO nanosquare irradiated with electromagnetic waves in normal glucose concentration in blood medium with a wavelength of (**a**) 450 nm, (**b**) 550 nm, and (**c**) 650 nm.

**Figure 8 molecules-28-00617-f008:**
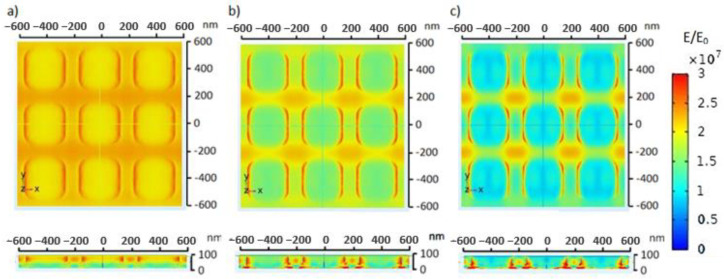
Top view and side view of the electric field distribution on the ZnO nanosquare irradiated with electromagnetic waves with a wavelength of 550 nm in (**a**) hypoglycemia, (**b**) normal, and (**c**) hyperglycemia blood media.

**Figure 9 molecules-28-00617-f009:**
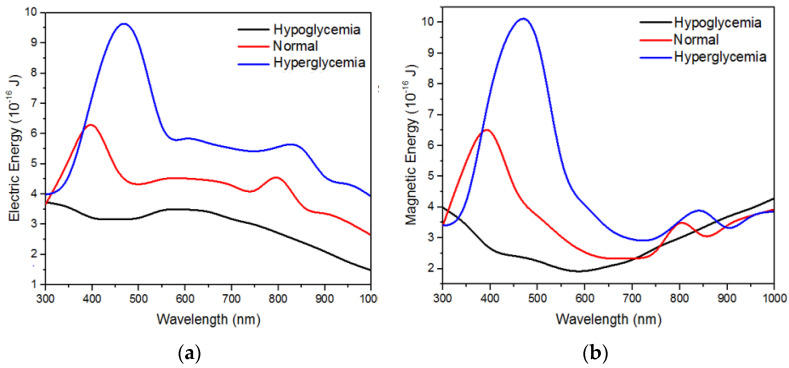
Simulation results of (**a**) electric energy and (**b**) magnetic energy on the ZnO nanosquare in hypoglycaemia (75 mg/dL), normal (100 mg/dL), and hyperglycemia (130 mg/dL) blood media.

**Figure 10 molecules-28-00617-f010:**
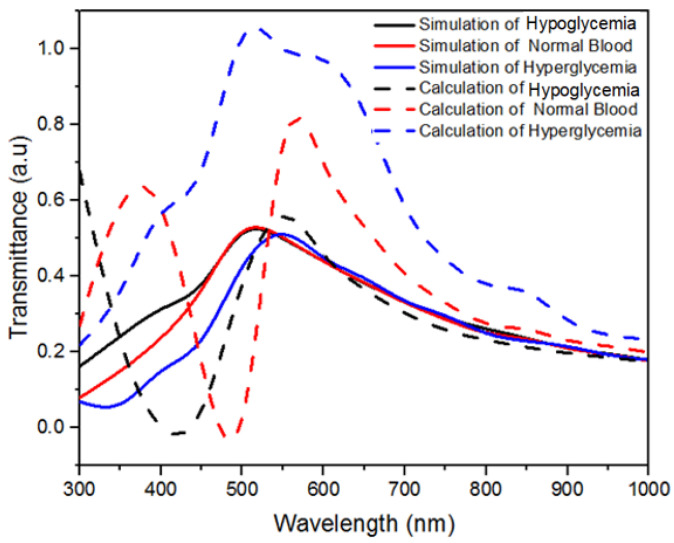
Transmittance of ZnO nanostructures—simulation and calculations results with variations of glucose concentration in blood media.

**Figure 11 molecules-28-00617-f011:**
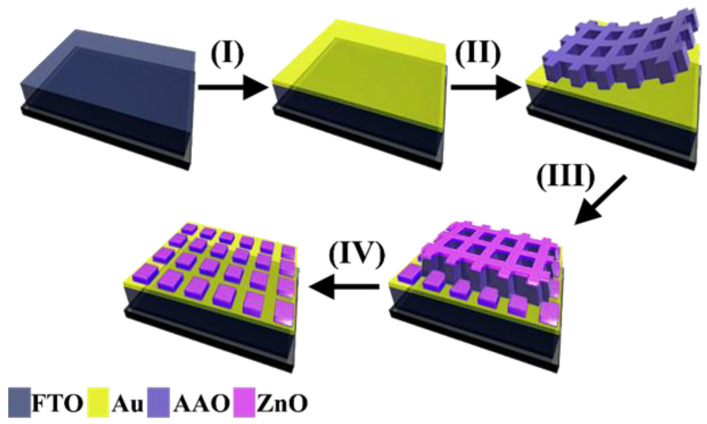
Illustration of ZnO/Au nanosquare array fabrication: (I) Deposition on a thin Au layer of around 5 nm, (II) Transferring AAO template onto FTO glass, (III) Deposition of ZnO, and (IV) removal of UTAM template.

**Table 1 molecules-28-00617-t001:** Comparison of the glucose sensor performance based on ZnO.

Electrodes	ZnO Structure	Sensitivity µA mM^−1^ cm^−2^	Reference
GOx/ZnO/Au nanosquare	Highly nanosquare array	188.34 mA mM^−1^ cm^−2^	This work
Nafion/GOx/ZnONFs/Au/PET	Urchin-like Nanostructure	0.57	[30]
Nafion/GOx/Fe3O4NPs/ZnONFs/Au/PET	Urchin-like Nanostructure	4.52	[30]
Au/ZnO	Nanorod array	24.56	[31]
Nafion/GOx/ZnO nanorods/ITO	Nanorod array	1.151 mA cm^−2^mM^−1^	[32]
GOx/ZnO/C/Paper	Nanowire	8.24	[33]

## Data Availability

Not applicable.

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
