# Peer review of "Highly Sensitive ZnO/Au Nanosquare Arrays Electrode for Glucose Biosensing by Electrochemical and Optical Detection"

_molecules, 2023, doi:10.3390/molecules28020617_

Round 1
Reviewer 1 Report
The article “Highly Sensitive ZnO/Au Nanosquare Array Electrode for Glucose Biosensing by Electrochemical and Optical Detection” demonstrated the fabrication of a ZnO/Au nanosquare array electrode and the detection of glucose concentration in biomedical applications.
Here are some questions for the authors:
1. Figure 3 and 5(b) and (d) correct the spelling of Oxidation in graph.
2. Figure 9(a) and (b) and Figure 10 correct the spelling of Hypoglycemia in graph.
3. Prepare a table showing the comparison of performance of the proposed platform over others for the glucose sensing.
4. Check the performance of the proposed sensor using physiological human serum/plasma/blood samples for the determination of glucose in real time.
5. Did the authors check the selectivity over other biomolecules Uric Acid or Ascorbic acid.
Author Response
Thank you for the review given. The comments given are very helpful for the development of our paper. We have made several changes to our paper which consist of the following points: (Anfortunetelly, some of Figure revision can not showing here. The reviewer evailable to see the attachment) below)
- Figure 3 and 5(b) and (d) correct the spelling of Oxidation in graph.
Response:
Thank you for the correction given. We have repaired the spelling of Oxidation in Figures 3(b), 3(d), 5(b) and 5(d).
Before
After
- Figure 9(a) and (b) and Figure 10 correct the spelling of Hypoglycemia in graph.
Response:
Thank you for the correction given. We have repaired the spelling of Hypoglycemia in Figures 3(b),3(d) 5(b) and 5(d).
Before
After
- Prepare a table showing the comparison of performance of the proposed platform over others for the glucose sensing.
Response:
Thank you for this correction. We have added a comparison table of job performance with existing research in line 313- line 320
“Table 1 shows the sensitivity value of glucose measurement compared to existing studies. Various forms of nanostructures were used in previous studies such as nanorods, nanowires, and urchin-like nanostructures. However, the results show that ZnO, which has a high nanostructure, is more sensitive than the others. In this work, obtained a high sensitivity value on glucose determination compared to previous research”
Tabel 1. Comparison of the glucose sensor performance based on ZnO
|
Electrodes |
ZnO Structure |
Sensitivity µA.mM-1.cm-2 |
Reference |
|
GOx/ZnO/Au nanosquare |
Highly nanosquare array |
188,34 mA mM-1 cm-2 |
This work |
|
Nafion/GOx/ZnONFs/Au/PET |
Urchin-like Nanostructure |
0.57 |
[29] |
|
Nafion/GOx/Fe3O4NPs/ZnONFs/Au/PET |
Urchin-like Nanostructure |
4.52 |
[29] |
|
Au/ZnO |
Nanorod array |
24.56 |
[30] |
|
Nafion/GOx/ZnO nanorods/ITO |
Nanorod array |
1.151 mA cm-2mM-1 |
[31] |
|
GOx/ZnO/C/Paper |
Nanowire |
8.24 |
[32] |
- Check the performance of the proposed sensor using physiological human serum/plasma/blood samples for the determination of glucose in real time.
Response:
CV testing on ZnO/Au nanosquare and ZnO/Au thin film is based on initial testing before it can be applied to real samples. However, based on the results obtained, this sensor can be applied to real samples in the future.
- Did the authors check the selectivity over other biomolecules Uric Acid or Ascorbic acid.
Response:
In this study, we used sucrose and fructose as other analytes in testing selectivity. This sensor proved to be selective in the presence of other analytes such as sucrose and fructose.
Thank you for the advice you gave. Hopefully the improvements we make can meet the publication criteria you want.
Best,
Nasori

Reviewer 2 Report
This work explained the fabrication of ZnO/Au thin film and ZnO/Au nanosquare array electrodes and conducted experiments to investigate their application in glucose biosensors. And investigated the quantum effect of the ZnO/Au thin film and nanosquare array electrodes in the presence of photon interactions on the nanostructure, the ZnO/Au nanosquare array achieved a higher sensitivity due to the larger surface-to-volume ratio compared to the ZnO/Au thin film, which have potential in optical applications such as light absorption by shooting photons for low blood glucose, normal blood glucose, and high blood glucose. Furthermore, finite difference time domain (FDTD) simulations and theoretical calculations on the energy density of the electric and magnetic fields produced by the ZnO/Au electrode were carried out. The proposed method possess some innovation for fabrication of the ZnO/Au nanosquare array, but some supplements are needed to enrich the article.
1. Why the performance of the ZnO/Au nanosquare array and thin film in detecting glucose concentration was electrochemically tested using CV in [Fe (CN) 6 ] 3-/4- and glucose mediums together (Fig.2)?
2. In the reduction and oxidation reaction that occurred in the CV test, the peak potential for the ZnO/Au thin film and ZnO/Au nanosquare was 0,078 V vs Ag/AgCl and 0,061 V vs Ag/AgCl (Fig.3). Who the potential (0,078 and 0.061 V) belongs to? How about signal of peak in range of 0.2-0.6 V ?
3.What is the difference between Fig.2 and Fig.5 presented results while both in [Fe (CN) 6 ] 3-/4- and glucose mediums together?
4.The larger increase in peak oxidation and reduction currents for the ZnO nanosquare is also caused by the larger surface area of the ZnO/Au nanosquare which contributes to the electrical conductivity for the electron transfer in flavin a denine dinucleotide (FDA) compounds and its derivatives such as FADH2. Where are the results indicating?
5. What are the signals of each peak of Fig.6b?
6. Where are the signals of Zn2+?
7. The serial number of the second level title is wrong in Materials and Methods.
Author Response
Thank you for the review given. The comments given are very helpful for the development of our paper. We have made several changes to our paper which consist of the following points:
- Why the performance of the ZnO/Au nanosquare array and thin film in detecting glucose concentration was electrochemically tested using CV in [Fe (CN) 6 ] 3-/4- and glucose mediums together (Fig.2)?
Response:
Potassium ferricyanide is used because it is a good mediator for the glucose sensor because of its cost efficiency and good electron transfer ability [2]. CV measurements were carried out in potassium ferricyanide medium with the addition of various glucose concentrations. This was done to determine the CV response to the addition of glucose and without glucose. From here it can also be known the sensitivity of the electrodes for normal conditions, low levels (hypoglycemia), and high levels (hyperglycemia). The use of Ferrocyanide medium and glucose simultaneously is adjusted to real samples. Where glucose detection is carried out on samples with various molecules, without further separation.
- In the reduction and oxidation reaction that occurred in the CV test, the peak potential for the ZnO/Au thin film and ZnO/Au nanosquare was 0,078 V vs Ag/AgCl and 0,061 V vs Ag/AgCl (Fig.3). Who the potential (0,078 and 0.061 V) belongs to? How about signal of peak in range of 0.2-0.6 V ?
Response:
Thank you for the correction given. I have corrected the information about the peak potential at 0.078 V for ZnO/Au thin film and 0.061 V for ZnO/Au nanosquare in line 170. CV testing carried out on ZnO/Au and ZnO/Au nanosquare thin films showed oxidation and reduction peaks in the range of 0.2-0.6 V. In the reduction and oxidation reaction that occurred in the CV test, the peak of oxidation potential for the ZnO/Au thin film and ZnO/Au nanosquare was 0.4 V vs Ag/AgCl and 0.55 V vs Ag/AgCl, respectively, at varying scan rates (from 0,12 Vs-1 up to 0,2 Vs-1 vs Ag/AgCl).
3.What is the difference between Fig.2 and Fig.5 presented results while both in [Fe (CN) 6 ] 3-/4- and glucose mediums together?
Response:
Thank you for the correction given. I have corrected the information in Figure 5. Figure 2 presents the results of CV measurements on [Fe(CN)6]3/4 and glucose media. While Figure 5 presents the results of CV measurements on media [Fe(CN)6]3-/4- and glucose with the addition of GOx (line 231).
4.The larger increase in peak oxidation and reduction currents for the ZnO nanosquare is also caused by the larger surface area of the ZnO/Au nanosquare which contributes to the electrical conductivity for the electron transfer in flavin a denine dinucleotide (FDA) compounds and its derivatives such as FADH2. Where are the results indicating?
Response:
The results that indicate the interaction of ZnO/Au nanosquare with FADH2 are shown in Figures 5b and 5d. This has also been explained in this paper, namely in line 267. "Furthermore, the increase in Ipca and Ipan as shown in Figures 5b and 5d indicates a faster electron transfer kinetics in the GOx active cycle. The increase in peaks is greater. Oxidation and reduction currents of ZnO nanosquare is also caused by the larger surface area of nanosquare ZnO/Au which provides electrical conductivity for electron transfer in flavin a denine dinucleotide (FDA) compounds and their derivatives such as FADH2. plays an active role as a catalyst in metabolism, where GOx reacts with FADH2 which has 2 charges negatively and 2 hydrogens are positively charged and produce more GOx and FADH2 which interact with the larger electrode surface”.
The improvement of sensor performance by nanostructures has also been analyzed by Qi mao.et al. Nanostructures with a high surface area can increase the contact area of glucose and increase the results of electron transfer which can be proven by the CV graph[3].
- What are the signals of each peak of Fig.6b?
Response :
Thank you for the additional discussion suggestions provided. The oxidation and reduction peaks at a potential of 0 V indicate that there is an oxidation and reduction reaction in glucose. While the peaks of oxidation and reduction at 0.4 V and -0.2 V indicate the presence of oxidation and reduction reactions in Zn. We have added citations and discussions regarding this information in the sentences on line 304.
“The oxidation and reduction peaks at a potential of 0 V indicate that there is an oxidation and reduction reaction in glucose. While the peaks of oxidation and reduction at 0.4 V and -0.2 V indicate the presence of oxidation and reduction reactions in Zn.“
- Where are the signals of Zn2+?
Response:
Thank you for the correction given. the standard reduction potential for Zn is -0.7 V. However, due to the influence of the pH of the solution, ZnO undergoes oxidation and reduction at a voltage of -0.4 V. we have already mentioned the reduction potential in line 304.
- The serial number of the second level title is wrong in Materials and Methods.
Response:
Thank you for the correction given. I have corrected the serial number in line 403, line 413, line 448, and line 461.
“3.1. Materials
3.2 Fabrication of ultra-thin alumina masks, Au thin films, and ZnO nanosquare arrays
3.3 Characterizations and Performance of Electrodes
3.4 Simulation Methods”
Thank you for the advice you gave. Hopefully the improvements we make can meet the publication criteria you want.
Best,
Nasori

Round 2
Reviewer 2 Report
No comments anymore.